



The distribution of sea-salt aerosol in the global troposphere

Daniel M. Murphy[1], Karl D. Froyd[1,2], Huisheng Bian[3,4], Charles A. Brock[1], Jack E. Dibb[5],
Joshua P. DiGangi[6], Glenn Diskin[6], Maximillian Dollner[7], Agnieszka Kupc[1,2,7], Eric M.
Scheuer[5], Gregory P. Schill[1,2], Bernadett Weinzierl[7,8], Christina J. Williamson[1,2], and Pengfei
Yu[1,2,9]

1. Chemical Sciences Division, NOAA Earth System Research Laboratory, Boulder, CO, USA

2. University of Colorado Cooperative Institute for Research in Environmental Sciences (CIRES)
at the NOAA Earth System Research Laboratory, Boulder, CO, USA

3. Joint Center for Environmental Technology University of Maryland Baltimore County,
Baltimore, MD, USA

4. Laboratory for Atmospheres, NASA Goddard Space Flight Center, Greenbelt, MD, USA

5. Earth Systems Research Center, Institute for the Study of Earth, Oceans, and Space,
University of New Hampshire, Durham, NH, USA

6. NASA Langley Research Center, Hampton, VA, USA

7. University of Vienna, Faculty of Physics, Boltzmanngasse 5, 1090 Vienna, Austria

8. Institut für Physik der Atmosphäre, Deutsches Zentrum für Luft- und Raumfahrt (DLR),
Oberpfaffenhofen, Germany

9. Institute for Environment and Climate Research, Jinan University, Guangzhou, China

Correspondence to: daniel.m.murphy@noaa.gov



**Abstract**

5    We present the first data on the concentration of sea-salt aerosol throughout most of the depth of
the troposphere and over a wide range of latitudes. Sea salt concentrations in the upper
troposphere are very small, usually less than 10 ng per standard m$^3$ (about 10 parts per trillion by
mass) and often less than 1 ng m$^{-3}$. This puts stringent limits on the contribution of sea-salt
aerosol to halogen and nitric acid chemistry in the upper troposphere. Within broad regions the

10    concentration of sea-salt aerosol is roughly proportional to water vapor, supporting a dominant
role for wet scavenging in removing sea-salt aerosol from the atmosphere. Concentrations of sea-
salt aerosol in the winter upper troposphere are not as low as in the summer and the tropics. This
is mostly a consequence of less wet scavenging in the drier, colder winter atmosphere. There is
also a source of sea-salt aerosol over pack ice that is distinct from that over open water. With a

15    well-studied and widely distributed source, sea-salt aerosol provides an excellent test of wet
scavenging and vertical transport of aerosols in chemical transport models.



# 1 Introduction

Sea salt particles are the largest aerosol component in the atmosphere by mass (Liao et al., 2006).

They represent about 30% of global column optical depth due to aerosols (Bellouin et al., 2013),
a somewhat smaller percentage than for mass because of their relatively large size compared to
other aerosols. Global climate models disagree on future changes in sea-salt aerosol due to
changes in wind speed and sea ice in a warming climate (Liao et al., 2006, Jones et al., 2007,
Boucher et al., 2013). Given the large contribution of sea salt to the global aerosol optical depth,

this could represent a significant climate feedback, with uncertainty even between different
scenarios in the same model (Hoose et al., 2008). Sulfuric acid, nitric acid, and some other acids
can displace halogens in salt particles. This provides both a sink for sulfate and nitrate and a
source of reactive chlorine, bromine, and iodine to the atmosphere (Chameides and Stelson,
1992; Finlayson-Pitts and Hemminger, 2000).

There is a large literature on the source of sea-salt aerosol as a function of wind speed (e.g.
Gong, 2003; Lewis and Schwartz, 2004; Grythe et al., 2014). There has been increased
recognition of the importance of submicron salt particles to aerosol number (Kreidenweis et al.,
1998; Clarke et al., 2003; Clarke et al., 2006). These submicron sea-salt particles are enriched in

organics compared to sea water, although the amount of enrichment is not consistent and may
vary with conditions (Middlebrook et al., 1998; Modini et al., 2010; Vignati et al., 2010;
Ovadnevaite et al., 2011; Gantt and Meskhidze, 2013).

Almost all of the sea-salt aerosol literature considers measurements within the marine boundary

layer and even there consists mostly of surface measurements. There have been few
measurements of how sea salt varies with altitude. Shinozuka et al. (2004) presented profiles of
non-volatile aerosol, presumed to be sea salt, up to about 2 km altitude for one region in the
Southern Ocean and one region in the tropical Pacific.

We present here the first measurements of the concentration of sea-salt aerosol over a wide range
of altitudes and latitudes. We consider the sea-salt vertical transport, wet removal, and



compositional variability. These data provide strong constraints on aerosol transport efficiency out of the MBL and are a useful tool in evaluating aerosol removal in large scale models.

## 2 Methods

We quantify sea-sea salt aerosol by merging measured size distributions with the fraction of particles in each size range identified as sea salt by single particle mass spectrometry. Sea-salt particles were identified from mass spectra of single aerosol particles using the Particle Analysis by Laser Mass Spectrometry (PALMS) instrument (Thomson et al., 2000). Particles enter a

vacuum and cross a split continuous laser beam. The transit time between the beams provides the particle velocity, used to determine its aerodynamic diameter. The aerosol inlet to PALMS is controlled to about 35 mbar, with a small dependence on outside pressure because the pressure transducer used for control was not positioned to capture the full effect of a jet downstream of the first critical orifice. Transit times were calibrated to known particle sizes at laboratory

pressure (about 820 mbar) before and after every field deployment. An excimer laser is triggered when a particle arrives at the second laser beam and ions are produced when the 193 nm pulse hits the particle. Either positive or negative ions are analyzed with a time-of-flight mass spectrometer, with the polarity switched every few minutes. For these data about 60% of the time was spent acquiring positive ion spectra.

The PALMS instrument was flown on the NASA DC-8 during the Atmospheric Tomography Mission (ATom) as well as earlier missions. The ATom mission consisted of several series of flights between about 85N and 65–80S latitude over both the Pacific and Atlantic Oceans. Flights consisted of successive en-route ascents and descents from about 160 m to 12 km with 5

to 15 minutes at the bottom and top of each profile. An extensive payload characterized both gas-phase and aerosol species (ATom, 2017: https://daac.ornl.gov/ATOM/guides/-ATom_merge.html). Two deployments are considered here: ATom1 from July 29 to August 22, 2016, in northern summer, and ATom2 January 26 to February 21, 2017, in northern winter. Some preliminary data from one ATom3 flight (October 14, 2017) are included to extend the

latitude range further south. During ATom1 and ATom2 PALMS acquired about 510,000 positive ion spectra and 350,000 negative ion spectra of individual particles.





Particle size distributions were measured by an Ultra-High Sensitivity Aerosol Spectrometer (UHSAS; Kupc et al., 2018) and a Laser Aerosol Spectrometer (LAS, TSI, Incorporated). UHSAS data were used for particles less than 0.51 µm diameter during ATom1 and 0.97 µm during ATom2; the LAS for larger particles. The two instruments agreed extremely well in the

overlap region (about 0.4 to 0.97 µm) for all of ATom1 and at low altitude during ATom2. A leak in the sheath flow of the LAS, traced to the threads of the set screws that center the inlet inside the sheath flow, led to the LAS under-sampling particles during ATom2 at high altitude when the aircraft cabin pressure was much higher than the sample pressure. The extra sheath flow reduced the sampling rate but did not introduce false counts, as checked occasionally in-

flight with filtered air. A correction was derived by comparing the LAS to the UHSAS, a Printed Optical Particle Spectrometer (POPS), and Cloud Aerosol Spectrometer (CAS) data.

The PALMS instrument has substantial biases in sampling efficiency for different size particles. In addition, a custom virtual impactor in the PALMS sampling line deliberately enriched the

concentration of super-micron particles in order to get better statistics for dust and sea salt. Overall, PALMS is much better at measuring fractional rather than absolute abundances of species such as sea salt. Combining such fractions with independently measured size distributions allows quantitative concentration measurements. Rather than directly calibrating the PALMS and virtual impactor sampling efficiencies, data here are normalized to the UHSAS and

LAS particle size distributions. Conceptually, if a given percentage of, for example, 1 to 2 µm diameter particles are sea salt, then the absolute concentration can be determined by multiplying the concentration of particles of that size from the LAS by the percentage of sea salt determined by PALMS. Although simple in concept, this normalization is complicated in detail because the UHSAS and LAS measure optical diameter, which depends on refractive index, whereas

PALMS measures aerodynamic diameter, which depends on density. The details of the normalization are given in Froyd et al. (in preparation).

PALMS, the LAS, and the UHSAS all sampled from a University of Hawaii inlet owned by NASA Langley on the DC-8 (McNaughton et al., 2007). About 1.5 m of 1/4" outside diameter

stainless steel tubing with a usual volumetric flow rate of 3.5 to 7 liters per minute led to a custom virtual impactor with a 1 µm cut point mounted on the PALMS instrument. The virtual




impactor design closely followed Loo and Cork (1988) except it was slightly scaled for our flow and desired cut point. After the virtual impactor about 20 cm of 1/8" stainless steel followed by 20 cm of 1/8" conductive Teflon tubing led to the PALMS focusing inlet. The volumetric flow after the virtual impactor was 0.7 liters per minute. The tubing was at aircraft cabin temperature

except that about the last 5 cm was taped to a heat pipe connected to the PALMS ion source region, which was temperature controlled at 35 °C. This slight warming to > 25 °C was done to avoid condensation in the aerodynamic focusing vacuum inlet on PALMS at low altitudes in the tropics. With the inlet tubing warmer than outside the aircraft the relatively humidity in the inlet was less than 40% for the majority of boundary layer sampling so the water content of the sea-

salt particles was reduced. Correlations between aerodynamic diameter and light scattering suggest that the salt particles did not effloresce in the inlet. Probably there was insufficient residence time for efflorescence even when the relative humidity in the inlet was very low. For sea-salt aerosol mass concentrations we assume that the sea-salt particles were deliquesced except if the outside relative humidity was less than 35%, when we assume they were dry. At

low temperatures sea salt partially effloresces at about 40% relative humidity (Koop et al., 2000).

The U. of Hawaii inlet on the DC8 has been shown to quantitatively transmit particles as large as 3.1 µm, at least at low altitude (McNaughton et al., 2007). Both the focusing inlet on PALMS and the LAS inlet tubing do not transmit particles larger than about 4 or 5 µm, so results here

represent particles smaller than about 3 µm (dry geometric diameter). Note that the measured size range of ~0.18 to 3 µm often represents a minority of sea-salt aerosol by mass, as indicated by the size distribution of large sea-salt particles detected by the cloud probes in the cloud-free marine boundary. When comparing to other data or model results, the size range must be considered.

In-cloud data are excluded from the results shown here. Huebert et al. (1990) showed that up to 90% of the largest sea-salt particles in the marine boundary layer can deposit to the walls of an inlet. Cloud droplets or ice crystals impacting a forward-facing aircraft inlet can act like a high-pressure washer to dislodge some of that salt, potentially leading to large sea-salt artifacts in

clouds. During both ATom and previous missions (Murphy et al., 2004), PALMS observed



anomalous particles in clouds, both sea-salt and other compositions, reinforcing the decision to exclude in-cloud data.

Sea-salt particles were identified in the positive ion mass spectra using a combination of peaks,
starting with a large $Na^+$ peak (greater than at least 17% or 30% of the ion current from a particle, depending on other peaks in the spectrum). Identification also required a potassium peak of appropriate size. Particles were excluded as sea salt if they contained aluminum (from clay minerals), barium (from fly ash and other minerals), or a variety of other metals. A hierarchical cluster analysis (Murphy et al., 2003) was also used. Although the cluster analysis generated
clusters of mass spectra that matched sea salt, criteria based on peak height were more reliable than the cluster analysis for identifying salt. Instead, the cluster analysis was most useful for excluding a few uncommon clusters that passed the peak height criteria but were not sea salt. For the great majority of sea-salt particles the identification was very clear. Figure 1 shows the mass spectrum of a single particle, chosen to be close to the average of all sea-salt mass spectra.

The main concern in identification is that at extremely low concentrations of sea-salt aerosol, as found over land or at high altitude, there may be a contribution from other particles, particularly dust from salt flats that is chemically similar to oceanic sea salt. A manual review was made of ~100 particles automatically classified as sea salt and a similar number automatically classified
as not sea salt. This was done for high-altitude coarse particles, the hardest region for the automatic classification. No definite sea-salt particles were missed and there was one definite false positive. More than 20 particles were borderline, mostly classified as sea salt by the automated algorithm. These were mostly spectra with sodium and potassium in a reasonable ratio for sea salt but magnesium at the wrong ratio. Note that if particles effloresce into
inhomogeneous crystals, then the PALMS laser can sometimes preferentially ionize just one region of a particle with varying amounts of magnesium. There was a much smaller percentage of borderline identification in the marine boundary layer where higher humidity led to mostly liquid particles that ionize more uniformly. The $Na^+$ peaks from sea-salt particles were often sufficiently large that they either saturated the detector or were broadened due to some sort of
repulsion in the ion source. This tended not to affect the identification of a mass spectrum as sea salt but only the quantification of the Na peak.





Sea-salt particles can also be identified from negative ion spectra using chloride ions and cluster ions containing Na. Data in this paper are from positive ion spectra because in aged sea-salt particles the chloride can be almost entirely displaced by sulfate and nitrate, making

identification more difficult with negative ions. In regions with fresh sea salt results from negative ion spectra were nearly identical to those from positive ion spectra.

The number and mass of sea-salt particles were calculated every 5 minutes of flight time in order to acquire enough mass spectra for a statistically significant normalization to the UHSAS and

LAS size distributions. During climbs and descents, 5 minutes represents about 2.5 km in altitude. Mass spectra of sea-salt particles were typically acquired at a rate of more than per second in the marine boundary layer and less than one per minute at high altitude.

Figure 2 shows concentrations of sea-salt aerosol at low altitudes measured by the PALMS/LAS

combination compared to filter measurements of sodium (Dibb et al., 1999). The filter samples indicate more sea-salt mass, which is expected because the inlet to the filter sampler transmitted larger particles than the inlet to PALMS. The good correlation adds confidence to the PALMS measurements in the upper troposphere, which are much more sensitive than the filter sampler. Because each particle is sampled at a particular time, its composition can be associated with a

particular altitude, aerosol concentration, water vapor concentration, and so forth. This allows for very long averaging times in similar air. For example, the average concentration of sea-salt aerosol in air with 10 to 20 ppmv of water can be calculated from collecting many such stretches of flight data even though they are not contiguous in time and might even be on different days. Furthermore, the PALMS single particle data are easier to screen for short periods of cloud and

other artifacts than are the extended filter samples. One can eliminate short cloud encounters without losing the data before and after the cloud. The internal consistency of the data suggests that the detection limit for sea salt is better than 10 ng m$^{-3}$ over a few minutes and better than 1 ng m$^{-3}$ when hours of data are available. Exact detection limits depend on the size distribution and the amount of internal mixing. In favorable circumstances the detection limits can be << 1

ng m$^{-3}$.



## 3 Results

The concentration of sea-salt aerosol in the marine boundary layer measured by PALMS was usually between 0.3 and 3 µg m⁻³. Such concentrations are reasonable considering the upper cut

point of about 3 µm diameter. The concentrations in the marine boundary layer were highly variable and modestly positively correlated with local wind speed (Shinozuka et al., 2004). Although the sea-salt aerosol production rate can be closely related to wind speed, the correlation with local wind speed is modest because the concentration also depends on the wind fetch and whether or not there has been recent precipitation. Sea-salt aerosol concentrations near the

surface were also correlated with relative humidity. This is expected since low relative humidity would indicate that the air was not well-mixed from the ocean surface up to the altitude of the DC-8. For example, during one boundary layer sampling leg over the ocean there was almost no sea salt at 160 m altitude but the relative humidity was much less than 50%, indicating limited surface influence for that particular leg.

There are already extensive measurements of sea-salt aerosol in the marine boundary layer (Lewis and Schwartz, 2004). The novel data here are the concentrations at higher altitudes. Figure 3 shows a latitude-altitude cross-section of sea-salt aerosol concentrations over the Pacific Ocean. Even though they were in different seasons, systematic differences between ATom1 and

ATom2 are not visible on a log color scale so they are combined in Figure 3. More subtle differences with season are discussed below. The Pacific Ocean is shown because there was less dust to complicate the analysis of very low concentrations; concentrations over the Atlantic Ocean were similar. Shinozuka et al. (2004) inferred sea-salt aerosol concentrations from non-volatile particles. The results shown here agree that this was valid for their measurements at < 2

km over the Pacific and Southern Oceans. However, it is not true in general that most non-volatile particles are sea salt. For example, at higher altitudes (e.g. 10 km) than Shinozuka et al. measured, non-volatile particles would be many times more likely to be dust than sea salt, even over the middle of the ocean. Dust concentrations measured during ATom will be discussed in future publications.


A salient property of the distribution of sea-salt aerosol in Figure 3 is that the concentration falls off rapidly with altitude, by about a factor of 10 for every 2 km. Above 6 km, the concentrations



were almost always less than 10 ng m$^{-3}$. The decrease at the top of the marine boundary layer was often very sharp in individual profiles. This was more apparent on past missions because the ATom mission ascents were relatively rapid. During the 2002 Intercontinental Transport and Chemical Transformation (ITCT) Experiment off the coast of California, PALMS was on the

NOAA P3 that often flew just above and below the top of the boundary layer. There, the sea-salt aerosol concentration could change by more than a factor of 10 in less than 100 m altitude at the top of the marine boundary layer.

Figure 4 shows size distributions of sea-salt particles from PALMS during ATom1. Most of the

sea-salt mass is in particles larger than 1 μm, most of the number is smaller than 1 μm. In the lower panel the size distribution from the upper troposphere is multiplied by 500 to get it on the same scale as the size distribution in the marine boundary layer, emphasizing the strong removal of sea salt. The number distribution is shown from the Southern Hemisphere because for submicron particles in the Northern Hemisphere both the number of particles and the fraction of

sea salt are very steep functions of diameter, leading to large uncertainties when they are multiplied. The mass distributions and the Southern Hemisphere number distribution are less steep. The high and low altitude size distributions (Figure 4b) show some differences of a factor of ~3 as a function of diameter. Such detailed differences in the size distribution with altitude vary by region. We would emphasize instead that an overall large removal factor is present

across the entire 0.3 to 3 μm diameter range. This may indicate that much of the removal of sea-salt aerosol was by nucleation scavenging in cloud rather than impaction by precipitation. Particles large enough to be measured by PALMS are all large enough to be condensation nuclei, whereas impaction scavenging is much more efficient for coarse particles than submicron particles particles (Pruppacher and Klett, 1997). The few sea-salt particles in the upper

troposphere were similar in size and more chemically processed than those in the marine boundary layer.

### 3.1 Sea-salt aerosol over pack ice

Sea-salt aerosols were not confined to areas with open water. Significant concentrations of sea-salt aerosol were also observed over ice-covered regions of the Arctic Ocean during ATom. For a portion of a flight north of Alaska on 19 February 2017, the nearest large areas of open water





were about 1000 km away. Yet significant concentrations of sea-salt aerosol (~ 1 µg m$^{-3}$ below 3 µm diameter) were observed at altitudes below 400 m. The sea-salt aerosol was only at low altitude, supporting a local source. Similar concentrations were also observed by PALMS on flights north of Alaska during the Aerosol, Radiation, and Cloud Processes affecting Arctic

Climate (ARCPAC) mission in March-April 2008. Significant sea-salt aerosol concentrations have also been measured during winter at ground-based Arctic stations (Quinn et al., 2002). Data from Barrow, Alaska, indicate some sea-salt aerosol production from leads but also show high concentrations of sea salt at times with no nearby open water in leads (May et al., 2016).

Mass spectra of sea-salt particles over the ice-covered Arctic Ocean were depleted in Na relative to Mg, K, and Ca as compared to particles at lower latitudes (Figure 5). The Na depletion over the Arctic was due to an increased population of particles with low Na rather than every particle having less Na. It is robust in the following sense: we used the Northern Hemisphere (NH) Pacific as a reference case for ATom2 because that is the region with the most data in the marine

boundary layer. We then compared data from other ocean regions with the NH Pacific. The magnitude of the difference in the average Na signal between the Arctic Ocean and the NH Pacific was much larger than the differences between the NH Pacific and any of the other open ocean regions, showing that the Arctic sea-salt aerosol is distinct from open ocean sea-salt aerosol.

Similar Na depletion was also observed during ARCPAC when comparing mass spectra of particles over Arctic Ocean compared to test flights over the Gulf of Mexico. In contrast, significant Na depletion over the Arctic Ocean was not observed during ATom1, which was flown during August when much of the Arctic Ocean had some open water. Unfortunately,

occasional detector saturation by the Na$^+$ peak as well as changing ionization patterns (such as the presence or absence of cluster ions) make it impossible to be more precise about the amount of Na depletion other than to say it was between about 20 and 50%.

Na depletion in polar sea-salt aerosol is consistent with Wagenbach et al. (1998) and Hara et al.

(2012), who found ≥10% depletion of Na relative to Mg and K in sea-salt aerosol presumed to be formed from sea ice around Antarctica. The PALMS data support the idea that some form of ice





brine, whether it be frost flowers and/or blowing briny snow, is an important source of sea-salt aerosol in the Arctic and Antarctic (Domine et al. 2004; Alvarez-Aviles et al, 2008; Huang and Jaeglé, 2017).

### 3.2 Sea-salt aerosol as a diagnostic for aerosol removal

Simple altitude profiles of PALMS sea-salt aerosol in various latitude bands will be presented in model-measurement comparison papers (Yu et al., 2018 submitted, Bian et al., in preparation, Zhang et al., in preparation). An alternative way of presenting the concentration of sea-salt aerosol is as a correlation with water vapor (Figure 5). Sea salt is water soluble, so one might expect that its removal would approximately scale with removal of water via precipitation. Figure 6 shows that this is the case, at least when considered as an average over many vertical profiles. Because the concentration of water vapor falls off rapidly with altitude, the correlations in Figure 6 are in a sense vertical profiles. However, the concentration of sea-salt aerosol is often better correlated with water vapor than with altitude itself.

The log-log slopes are not far from one, indicating that sea-salt aerosol is removed with water: when 90% of the water is removed about 90% of the sea-salt aerosol is removed. Labels indicate water mixing ratios beyond which most clouds are ice or most are liquid water, based on saturation vapor pressures of -15 to -5 C at 850 to 500 mbar. There is some indication, especially in the Northern Hemisphere data, that sea-salt aerosol removal is more efficient in liquid water clouds than in ice clouds. The log slopes are greater than 1 at high water mixing ratios and close to or less than 1 at low water mixing ratios. Some proportionality between sea-salt aerosol and water vapor continues to very low water mixing ratios where the clouds must be ice rather than liquid water. This is somewhat surprising, since the Bergeron process whereby ice crystals grow at the expense of water droplets in a mixed phase cloud could leave many salt particles behind even if those particles had originally acted as condensation nuclei. There are several possible explanations for the continued removal of sea-salt aerosol at very low temperatures. Sodium chloride dihydrate can be an effective ice nucleus below 227 K K (Wagner and Möhler, 2013). Non-spherical ice crystals can scavenge aerosols by impaction more efficiently than spherical droplets (Chapter 17, Pruppacher and Klett, 1997) A less microphysical explanation could be that the driest air in the upper troposphere is the result of very deep convection. If such intense



convective clouds scavenge nearly all sea-salt particles, then the observed correlation of sea-salt aerosol with water vapor at low concentrations could be due to mixing very dry air containing near-zero sea-salt aerosol with mid-tropospheric air containing more of both water and salt.

In many ways Figure 6 shows a remarkably simple picture of sea-salt aerosol concentrations by season and hemisphere. Since the ATom1 and ATom2 deployments were roughly six months apart and covered both hemispheres, it is possible to distinguish seasonal and hemispheric differences. Concentrations in the tropical atmosphere show little seasonal dependence. The two summer hemispheres are fairly similar to the tropics. The two winter curves are shifted up and to
the left. Especially in the Northern Hemisphere, the sea-salt concentration near the ocean surface (at the top right of each curve) is not all that different in winter and summer. Instead, a similar amount of sea-salt aerosol is emitted into a lower absolute humidity in the colder winter air. This suggests two reasons more sea-salt aerosol can reach the upper troposphere in winter than in summer. The main reason is that more sea-salt particles can survive into the upper troposphere in
winter simply because in winter there is less water available to wash out the aerosol. Second, removal of sea-salt aerosol in ice clouds may be less efficient than in liquid water clouds.

Model results for the correlation between sea-salt aerosol and water vapor are shown in Figure 7. For simplicity only a subset of the regions in Figure 6 are shown. CESM-CARMA couples a
sectional aerosol model (Yu et al., 2015; Toon et al., 1988) with the National Science Foundation/Department of Energy Community Earth System Model (CESM). CARMA uses 20 size bins for sea spray aerosols which are composed of salt, marine sulfate, and marine organics. The GEOS5 model simulates meteorological fields to drive online GOCART aerosol module (Colarco et al., 2010; Bian et al., 2013). GOCART sea-salt aerosol is emitted using an upgraded
emission algorithm (Gong et al., 2003; Bian et al., in preparation) and removed by warm cloud from convective updraft and large-scale rainout and washout, as well as by dry deposition and sedimentation (Chin et al., 2002). A humidified sea-salt particle size (Gerber, 1985) is used for computations of particle fall velocity, deposition velocity, and optical parameters. The detailed description of the GEOS5-GOCART sea-salt aerosol simulation for this work is given in Bian et
al., (in preparation).



Both models reproduce the strong correlation between sea-salt aerosol and water vapor. Both models also capture the difference between the winter and summer hemispheres in the correlation. The GEOS5 model may be removing aerosol somewhat too slowly in ice clouds. The comparison to these data uncovered an error in aerosol removal in the CESM model in which

sea-salt aerosol was originally overestimated by a factor of over 100 in the upper troposphere. One example is shown in Figure 7. The overestimate was traced to aerosols not being properly removed from air transported in the sub-grid convective parameterization. A detailed analysis and new removal parameterization are described by Yu et al. (submitted). An interaction of removal and entrainment parameterizations was also identified as an issue in the CAM5 model

by Wang et al. (2013) based on black carbon data in the upper troposphere. The CARMA bin microphysics also reproduces the similar shape of the size distribution of sea-salt aerosol at different altitudes (Figure 4b). In the model, only particles larger than about 3 μm decrease strongly with altitude due to sedimentation.

**3.3 Implications for reactive gases**

One consequence of the small concentrations of sea-salt aerosol in the upper troposphere is that it contributes little to chemical reactivity. In particular, Wang et al. (2015) postulated an important role for upward transport of sea-salt aerosol followed by release of bromine into the

upper troposphere. These data show that debromination of sea salt cannot be a significant source in the upper troposphere. There was almost always less than 10 ppt of sea-salt aerosol by mass and often less than 1 ppt. Given that sea salt is very roughly 0.1% by mole bromine, there would be parts per quadrillion of bromine available from transported sea-salt aerosol. If bromine from sea-salt aerosol is to significantly affect the upper troposphere it would have to be released at low

altitude and transported in the gas phase, although it is not clear if there are any suitable gas-phase bromine compounds that would survive wet scavenging.

The small concentrations of sea-salt aerosol in the upper troposphere also provide a strong constraint on the influence of salt on the gas phase reactive nitrogen budget. Even complete

replacement at altitude of sea-salt chlorine by nitrate would be a very small sink for nitrate compared to the hundreds of pptv of $NO_y$ in the upper troposphere (Weinheimer et al., 1994; Emmons et al., 1997). $NO_y$ measurements are available as part of the ATom data set (ATom,





2017). A similar argument applies to sulfate: in the upper troposphere the maximum amount of sulfate in sea-salt aerosol is far less than the amount of sulfate in mixed sulfate-organic particles. Note that the minimal chemical importance of sea-salt aerosol is for the upper troposphere only: in the marine boundary layer reactions with sea-salt particles can significantly modify the gas

phase bromine, nitrate, and sulfate budgets.

## 4 Summary

These are the first measurements of sea-salt aerosol over a wide range of altitudes and latitudes.

Data are available from near the surface to about 12 km altitude from about 65S to 80N. The final ATom data set will add two more seasons and extend the data to beyond 80S. Detailed comparisons to chemical transport models are underway (Bian et al., in preparation, Zhang et al., in preparation). One of these comparisons identified a problem in aerosol vertical transport in the CESM model (Yu et al., submitted).

Sea-salt aerosol has only a surface source and a sink by scavenging (i.e. sea-salt particles do not evaporate). That makes sea-salt aerosol a powerful tool to study wet removal of aerosol. The data here indicate that removal of sea-salt aerosol is very approximately proportional to the removal of water over a wide range of absolute humidity, with possibly more efficient removal in liquid

water clouds than in ice clouds.

**Acknowledgments.** PALMS participation in the Atmospheric Tomography flights (ATom1 and ATom2) was supported by NOAA climate funding. The mission as a whole was supported by NASA's Earth System Science Pathfinder Program EVS2 funding. C. Brock and C. Williamson

were supported by award NNH15AB12I and by NOAA's Health of the Atmosphere and Atmospheric Chemistry, Carbon Cycle, and Climate Programs. A. Kupc was supported by the Austrian Science Fund FWF's Erwin Schrodinger Fellowship J-3613. The CESM project is supported by the National Science Foundation and the Office of Science (BER) of the US Department of Energy. We thank the ATom team and crews for making the flights possible, and

B. Anderson of NASA Langley for the use of the U. Hawaii inlet. Data are available at https://dx.doi.org/10.5067/Aircraft/ATom/TraceGas_Aerosol_Global_Distribution and http://esrl.noaa.gov/csd/projects/atom/data.php.




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

**Figure 1:** Mass spectra of (a) a typical sea salt particle and (b) a highly processed sea salt particle where the chloride has been replaced by nitrate and sulfate. The spectrum in (a) is chosen to be similar to the average of all low-altitude spectra of sea salt.



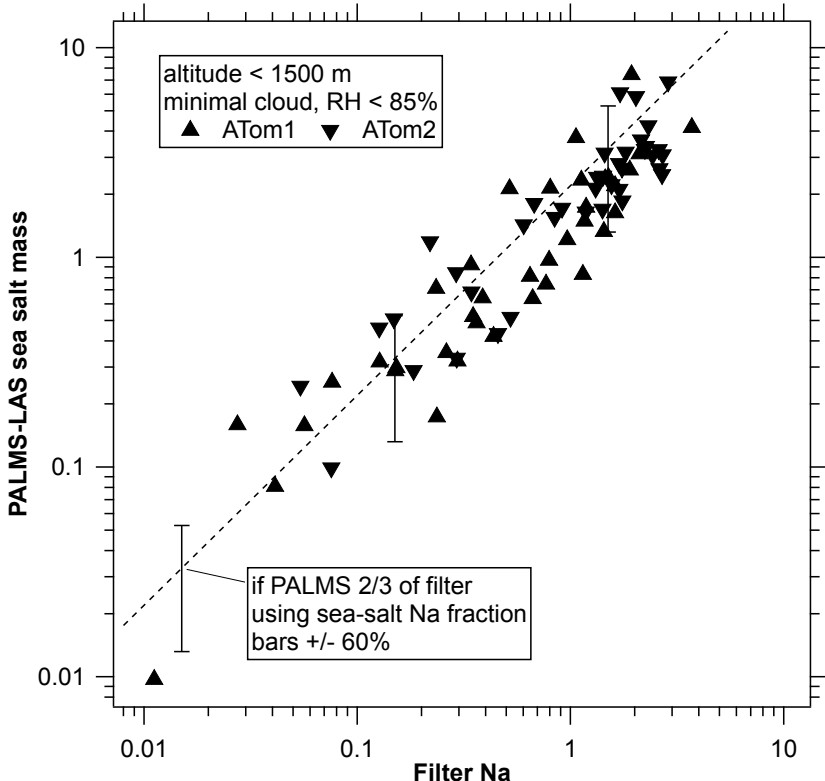

**Figure 2:** Sea-salt aerosol mass at low altitudes derived from PALMS compared to filter samples
of aerosol sodium. The inlet for the filter sampler samples larger particles than the PALMS inlet,
hence PALMS is expected to sample somewhat less mass. The cutoff at 85% relative humidity is
imposed because comparing the different inlet cut points becomes especially problematic when
the particles are enlarged due to water uptake at high humidity.





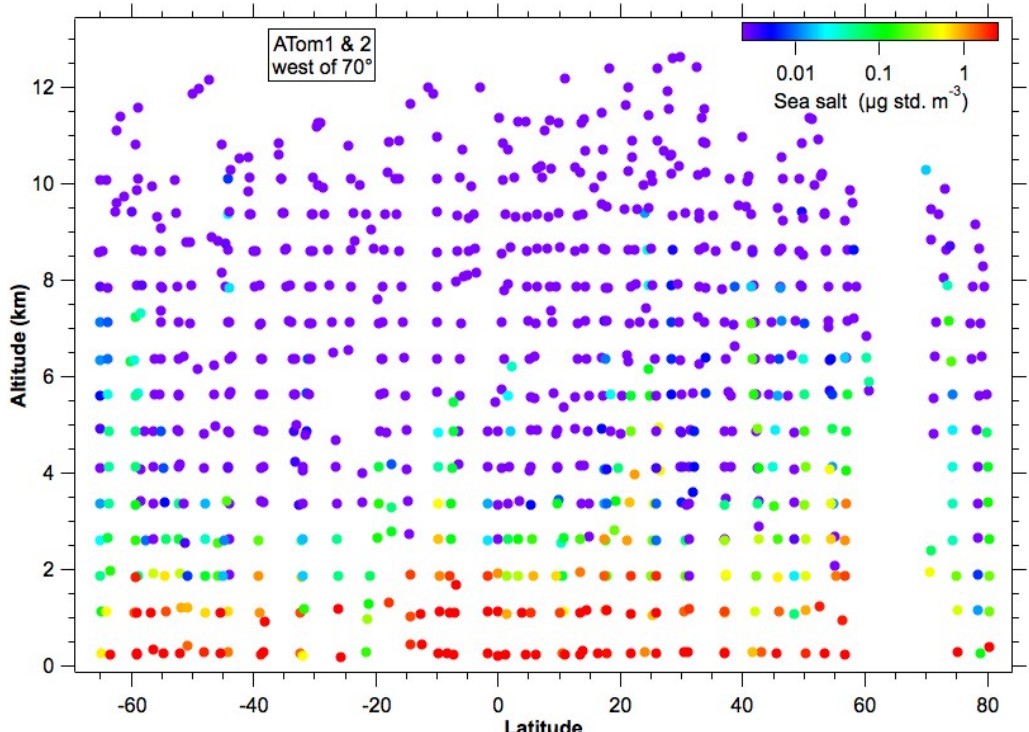

**Figure 3:** Concentration of sea-salt aerosol during the ATom1 and ATom2 flights over the Arctic, Pacific, and Southern oceans. The blank region between about 60 to 70N is because of flying over land (Alaska). The color scale is from 2.5 ng m⁻³ to 2.5 µg m⁻³ (at standard conditions). These concentrations include sea-salt particles between about 180 nm and 3 µm diameter. Data are averaged over bins of 750 m vertically and about 4 degrees latitude; points are plotted at the midpoint location of the data within each bin.





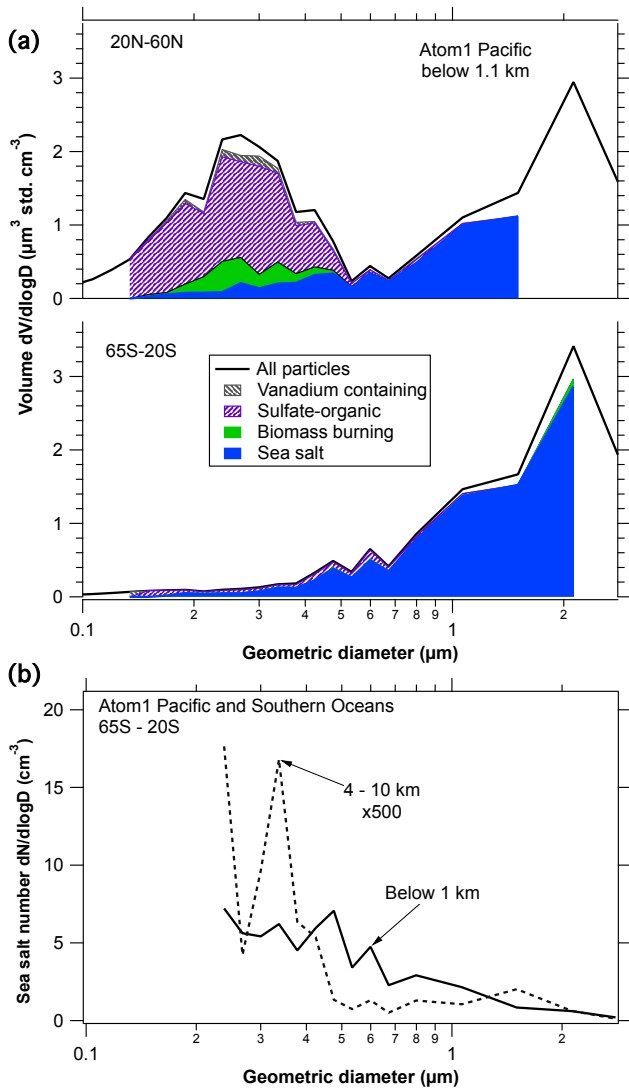

**Figure 4:** Size distributions of sea-salt particles in the marine boundary layer. The upper panel shows how the total size distribution (black line) is multiplied by PALMS composition at each size to obtain the volume of sea-salt particles. White areas below the size distribution curves represent unknown compositions or, at the largest and smallest diameters, insufficient data to assign composition. The reduced concentration above about 3 μm diameter is because the aircraft inlet does not efficiently transmit larger particles. The lower panel compares the number size distributions at low and high altitudes.



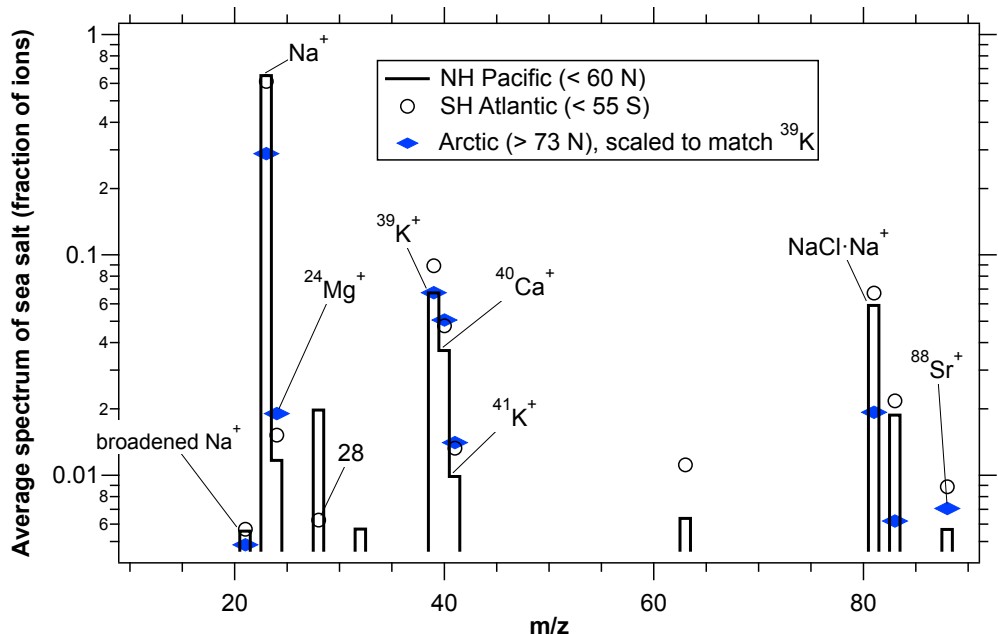

**Figure 5:** Average spectra of sea-salt particles larger than 1 μm diameter for the Arctic and lower latitude regions during ATom2. Na peaks were smaller over the Arctic Ocean than over the Northern Hemisphere Pacific Ocean. The South Atlantic is included to provide an estimate of consistency between regions that should be similar. Spectra were selected with ion intensities large enough to measure minor peaks but not significantly saturate the Na peak. n≈5500 for the NH Pacific, 700 for the SH Atlantic, and 300 for the Arctic.





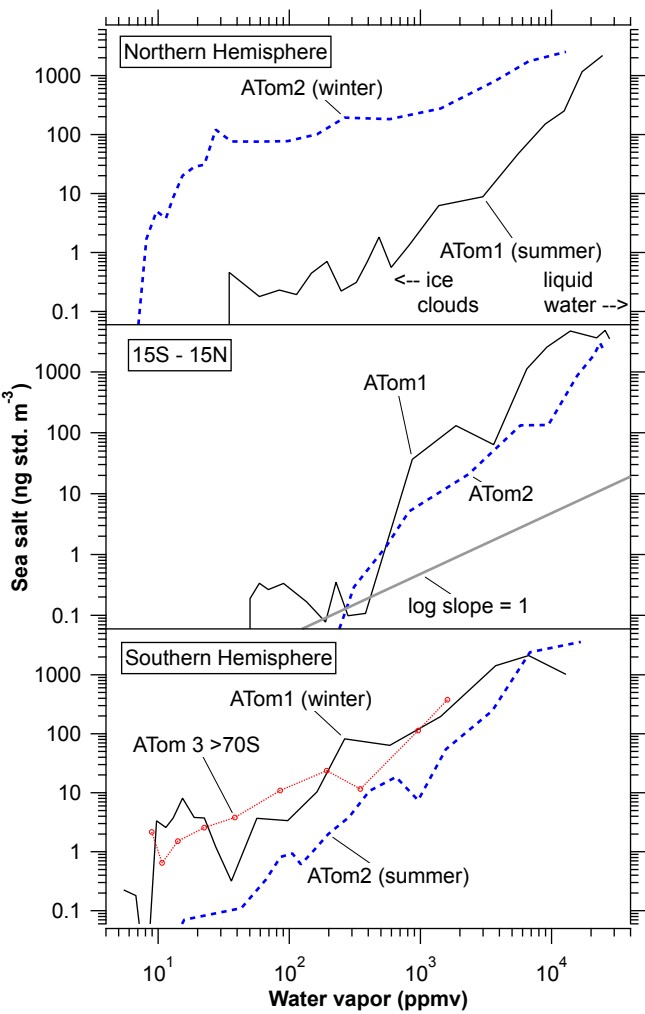

**Figure 6:** Measured sea-salt mass concentrations as a function of water vapor. Latitude bands for
the Northern and Southern Hemispheres are 20 to 65° except for the labeled data over Antarctica.
All data except that beyond 70S are from the Pacific Ocean side of North and South America.
Vertical lines show very approximate water mixing ratios above which most clouds are liquid
water and below which most clouds are ice. One way to view the graph is to consider water
vapor as a vertical scale with wet air at low altitudes and dry air at high altitudes. In both
hemispheres the winter data show more sea-salt aerosol in the upper troposphere than either the
summer hemisphere or the tropics.





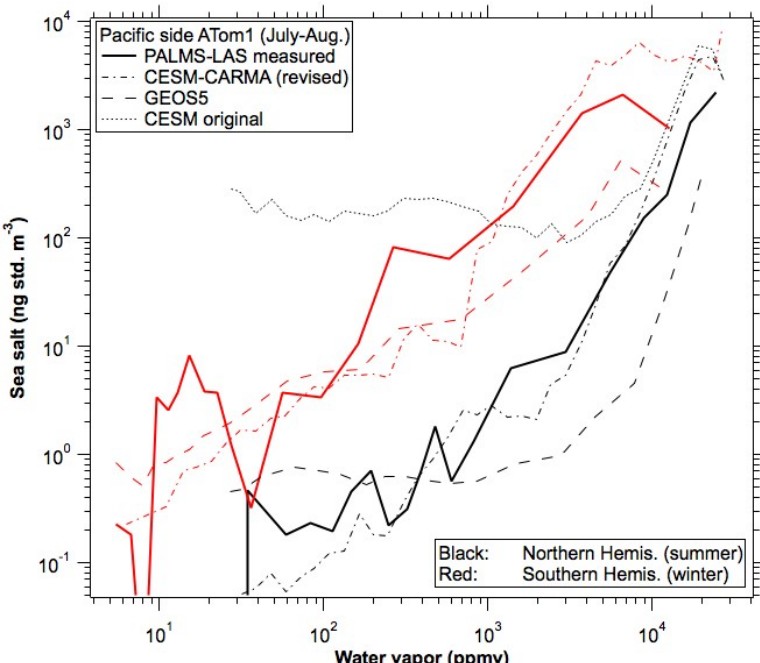

**Figure 7:** Model correlations for the CESM-CARMA and GEOS5 models. ATom2 and tropical curves are omitted for simplicity. GEOS5 model output was sampled along the flight tracks and CESM-CARMA at all altitudes in a curtain along the flight tracks. Also shown is one curve from the CESM-CARMA model before a revised convective aerosol removal scheme was implemented (Yu et al., submitted).