# Peer review of "The distribution of sea-salt aerosol in the global troposphere"

_Atmospheric Chemistry and Physics, 2018_

## Referee Comment (RC1) · Anonymous Referee #1 · 2 Dec 2018

The authors present observations of sea-salt aerosol concentrations collected with the Particle Analysis by Laser Spectrometry (PALMS) instrument on the ATom aircraft campaign in July-August 2016 and January-February 2017 as well as one flight in October 2017. These observations afford a global view of sea salt aerosol size distribution over a large range of latitudes (85N-80S) and altitudes (surface to 12 km). The measured diameter range is from 0.18 to 3 micrometers. The authors find a strong altitude dependence of the sea salt concentrations and a strong correlation with water vapor, reflecting wet scavenging of sea salt in the atmosphere. The authors report a source of sea salt aerosol over sea ice, with a different chemical composition than over the open ocean. The authors also compare these observations with results from chemical transport models.

[Figure]

The paper presents a very unique and interesting dataset of sea salt aerosol mass concentrations. The uniqueness comes from the systematic observations with the same instrument over a large of altitudes, latitudes, and for different seasons. The study is within the scope of ACP and represents a new contribution to the field of sea salt aerosol spatial distribution and scavenging. The paper is well written and well organized. I have a few comments that I would like to see the authors address.

1) Page 5 line 26. The authors mention that the details of the normalization are provided in a manuscript that's in preparation. It would be useful to include a bit more detail on this normalization in the present manuscript. For example, how large are the normalization factors that are applied to the PALMS instrument? How do these factors vary with particle size? Is there a dependence on altitude?

2) Figure 2. The text states that the filter samples indicate more sea-salt mass, as expected from sampling larger particles than PALMS. However, when looking at the figure it looks like PALMS is systematically larger than the filter Na+ measurements: nearly all the points appears to fall above the 1:1 line and PALMS observations of sea salt mass are larger than Filter Na+. Am I misreading the chart? It would be useful to give the statistics of the slope and correlation coefficient associated with the dashed line on the plot. Also, the figure has a box "if PALMS 2/3 of filter..." This statement is unclear, can the authors please explain in the figure legend that the error bars correspond to?

3) The distinct composition of sea salt aerosol over Arctic sea ice is an interesting result, consistent with a sea ice origin. Can the authors elaborate on the size distribution of these sea salt particles? Does it also look distinct from open ocean sea salt? For example, if these sea salt particles originate from frost flowers, one might expect larger particles. If they originate from blowing snow, then these is a possibility that submicron particles would be present, depending on the size of the snow particles and the numbers of sea salt particles produced per snowflake that sublimates.

4) The authors interpret the sea salt –water vapor correlation plots in Figure 6 as indicators of scavenging of water vapor and sea salt, and effectively vertical profiles of sea salt. However, one confounding factor in terms of the seasonal differences seem between ATom1 (summer) and Atom2 (winter) is the different temperature profiles. Based on Clausius-Clapeyron, I assume that a similar water vapor mixing ratio – say 1000 ppmv – corresponds to different altitudes for winter and summer in the northern hemisphere as well as for the tropics (15S-15N panel) as the temperature profiles are likely quite different. To provide further support the statement "In both hemispheres the winter data show more sea-salt aerosol in the upper troposphere than either the summer hemisphere or the tropics.", the authors would have to plot sea salt concentrations as a function of latitude for different seasons above a certain altitude. Alternatively, the authors could show the mean or median vertical profiles of sea salt aerosol mass concentrations for different seasons and latitude bands. Are the values shown in Figure 6 means or medians of sea salt concentrations for each water vapor bin? Having a sense of the variability for the blue and black lines in the various panels of Figure 6 would be useful. This could be done by showing an envelope of the sea salt concentrations (25th and 75th percentile for example).

Additional minor comments

- page 8 line 11-12. Is "one" missing from "a rage of more than per second"?

- page 10 line24. Remove extra "particles"

- page 12. Line 10. Do the authors mean Figure 6 instead of 5?

---

## Referee Comment (RC2) · Anonymous Referee #2 · 14 Dec 2018

ACPD Manuscript "" The distribution of sea salt aerosol in the global troposphere", by D.M. Murphy et al.

The manuscript describes results of sea salt aerosol measurements from the ATom aircraft campaigns. The ATom campaigns provided unique datasets on many aspects of the properties and composition of the Troposphere that are particularly well suited for model evaluation purposes. Systematic measurements of sea salt particle concentrations in the Troposphere are rare. Therefore this dataset provides important information about the 3D distribution of this aerosol type. The authors speculate about the reasons for particular distributions within the observed vertical sea salt concentration profiles. The manuscript is well written and the results are useful. However, some comments should be taken into account by the authors before this manuscript should be considered for publication in ACP.

A large part of the manuscript deals with the technical aspects of the measurements. I cannot comment on the methods, as this topic is beyond my expertise. Instead my remarks are about the science and interpretation of the measurement data.

Specific remarks:

- The ATom campaign should already be mentioned in the Abstract.

- The presentation of the results in the manuscript is mostly descriptive; at some places more detail would be helpful. The interpretations of the measurements remain quite speculative. E.g., on page 10 of the manuscript, the authors connect lesser large particle concentration in the upper troposphere to washout processes. This assumption is not supported by additional information – e.g. is there reason to assume gravitational settling cannot also play a role? Also, would cloud drying not lead to smaller particle sizes?

- Page 9, top: The authors describe the correlation of boundary layer sea salt with local wind speeds. For which height is the comparison performed? The authors claim that these variables are only weakly correlated, but do not provide any numbers or a figure to support this finding. Instead, they reference another earlier publication unrelated to the ATom measurements. Please clarify if this lack of correlation also is valid for the ATom data, and, if that is the case, a figure or correlation numbers would be appreciated. The weak correlation can be interpreted due to other influence factors controlling the sea salt concentrations such as relative that is mentioned here. However these are not shown either.

- Page 11: Are any correlations between sea salt aerosol and surface winds or humidity found over land?

- In subsection 3.2 the sea salt correlation with humidity is taken as indicator for wet removal of the particles. In contrast, on page 9 the correlation is is explained as indicator

for ‚ixing of air masses. Please clarify.

- The authors refer to model results shown in Figure 7 partly from submitted or 'in preparation' publications. This should be avoided.

---

## Author Comment (AC1) · 5 Feb 2019

We thank the reviewers. Specific responses to the reviews follow:

Anonymous Referee #1

The authors present observations of sea-salt aerosol concentrations collected with the Particle Analysis by Laser Spectrometry (PALMS) instrument on the ATom aircraft campaign in July-August 2016 and January-February 2017 as well as one flight in October 2017. These observations afford a global view of sea salt aerosol size distribution over a large range of latitudes (85N-80S) and altitudes (surface to 12 km). The measured diameter range is from 0.18 to 3 micrometers. The authors find a strong altitude dependence of the sea salt concentrations and a strong correlation with water vapor,

reflecting wet scavenging of sea salt in the atmosphere. The authors report a source of sea salt aerosol over sea ice, with a different chemical composition than over the open ocean. The authors also compare these observations with results from chemical transport models.

The paper presents a very unique and interesting dataset of sea salt aerosol mass con- centrations. The uniqueness comes from the systematic observations with the same instrument over a large of altitudes, latitudes, and for different seasons. The study is within the scope of ACP and represents a new contribution to the field of sea salt aerosol spatial distribution and scavenging. The paper is well written and well organized. I have a few comments that I would like to see the authors address.

1) Page 5 line 26. The authors mention that the details of the normalization are provided in a manuscript that's in preparation. It would be useful to include a bit more detail on this normalization in the present manuscript. For example, how large are the normalization factors that are applied to the PALMS instrument? How do these factors vary with particle size? Is there a dependence on altitude?

**More details have been added to this paragraph. **

2) Figure 2. The text states that the filter samples indicate more sea-salt mass, as expected from sampling larger particles than PALMS. However, when looking at the figure it looks like PALMS is systematically larger than the filter Na+ measurements: nearly all the points appears to fall above the 1:1 line and PALMS observations of sea salt mass are larger than Filter Na+. Am I misreading the chart? It would be useful to give the statistics of the slope and correlation coefficient associated with the dashed line on the plot. Also, the figure has a box "if PALMS 2/3 of filter..." This statement is unclear, can the authors please explain in the figure legend that the error bars correspond to?

**We agree the figure was confusing and it has been revised. The 1:1 line is displaced because not all sea-salt mass is sodium.**

3) The distinct composition of sea salt aerosol over Arctic sea ice is an interesting result, consistent with a sea ice origin. Can the authors elaborate on the size distribution of these sea salt particles? Does it also look distinct from open ocean sea salt? For example, if these sea salt particles originate from frost flowers, one might expect larger particles. If they originate from blowing snow, then these is a possibility that submicron particles would be present, depending on the size of the snow particles and the numbers of sea salt particles produced per snowflake that sublimates.

\*\*This is a good question. We looked back at the data for this reviewer question and the size distributions are somewhat different. However, for various technical reasons it is hard to pull out a good quantitative size distribution of the sodium-depleted particles and we are more comfortable with a qualitative statement. Therefore, the following has been added to section 3.1: "During both ARCPAC and ATom, sea-salt particles were also somewhat smaller over the frozen Arctic Ocean than other regions." \*\*

4) The authors interpret the sea salt –water vapor correlation plots in Figure 6 as indicators of scavenging of water vapor and sea salt, and effectively vertical profiles of sea salt. However, one confounding factor in terms of the seasonal differences seem be- tween ATom1 (summer) and Atom2 (winter) is the different temperature profiles. Based on Clausius-Clapeyron, I assume that a similar water vapor mixing ratio – say 1000 ppmv – corresponds to different altitudes for winter and summer in the northern hemisphere as well as for the tropics (15S-15N panel) as the temperature profiles are likely quite different. To provide further support the statement "In both hemispheres the winter data show more sea-salt aerosol in the upper troposphere than either the summer hemisphere or the tropics.", the authors would have to plot sea salt concentrations as a function of latitude for different seasons above a certain altitude. Alternatively, the authors could show the mean or median vertical profiles of sea salt aerosol mass concentrations for different seasons and latitude bands. Are the values shown in Figure 6 means or medians of sea salt concentrations for each water vapor bin? Having a sense of the variability for the blue and black lines in the various panels of Figure 6 would be

useful. This could be done by showing an envelope of the sea salt concentrations (25th and 75th percentile for example).

**The reviewer is correct that different temperature profiles can complicate the winter-summer comparison. We have clarified the caption of Figure 6 to say that "the winter data show more sea-salt aerosol in the upper troposphere at a given amount of water vapor". At some altitudes the winter hemisphere has more in an absolute sense as well, but the main point is that the winter hemisphere has more when normalized to water vapor. The caption has been changed to indicate average concentrations. Note that a manuscript (Bian et al.) with a model-measurement comparison of altitude profiles is now in discussion at ACPD.

Adding variability to Figure 6 is difficult because at the low concentrations in the upper troposphere much of the short-term variability in sea salt is due to statistical fluctuations in particles entering the instrument rather than real atmospheric variability. Figure 3 conveys some of the variability.**

Additional minor comment  - page 8 line 11-12. Is "one" missing from "a rage of more than per second"? **Fixed** - page 10 line24. Remove extra "particles" **Fixed** - page 12. Line 10. Do the authors mean Figure 6 instead of 5? **Fixed**   Interactive comment on Atmos. Chem. Phys. Discuss., https://doi.org/10.5194/acp-2018-1013, 2018. Anonymous Referee #2

 ACPD Manuscript "" The distribution of sea salt aerosol in the global troposphere", by D.M. Murphy et al. The manuscript describes results of sea salt aerosol measurements from the ATom aircraft campaigns. The ATom campaigns provided unique datasets on many aspects of the properties and composition of the Troposphere that are particularly well suited for model evaluation purposes. Systematic measurements of sea salt particle concentrations in the Troposphere are rare. Therefore this dataset provides important information about the 3D distribution of this aerosol type. The authors speculate about the reasons for particular distributions within the observed vertical sea salt concentration profiles. The manuscript is well written and the results are useful. However, some comments should be taken into account by the authors before this manuscript should be considered for publication in ACP. A large part of the manuscript deals with the technical aspects of the measurements. I cannot comment on the methods, as this topic is beyond my expertise. Instead my remarks are about the science and interpretation of the measurement data.

Specific remarks: - The ATom campaign should already be mentioned in the Abstract. **Done. **

- The presentation of the results in the manuscript is mostly descriptive; at some places more detail would be helpful. The interpretations of the measurements remain quite speculative. E.g., on page 10 of the manuscript, the authors connect lesser large particle concentration in the upper troposphere to washout processes. This assumption is not supported by additional information – e.g. is there reason to assume gravitational settling cannot also play a role? Also, would cloud drying not lead to smaller particle sizes?

**We have clarified the statement about sedimentation (end of section 3.2) to mention gravitation. We do not understand the comment about cloud drying leading to smaller particle sizes. If cloud droplets form and evaporate components such as sodium are the same as before the droplets formed and secondary components such as sulfate are usually larger. **

- Page 9, top: The authors describe the correlation of boundary layer sea salt with local wind speeds. For which height is the comparison performed? The authors claim that these variables are only weakly correlated, but do not provide any numbers or a figure to support this finding. Instead, they reference another earlier publication unrelated to the ATom measurements. Please clarify if this lack of correlation also is valid for the ATom data, and, if that is the case, a figure or correlation numbers would

be appreciated. The weak correlation can be interpreted due to other influence factors controlling the sea salt concentrations such as relative that is mentioned here. However these are not shown either.

\*\*The sentence has been clarified that the wind speed referred to was at flight altitude and that modest correlations between sea salt and wind speed were found both for these data and the earlier publication. \*\*

- Page 11: Are any correlations between sea salt aerosol and surface winds or humidity found over land?

\*\*We have not examined this. Sea salt aerosol concentrations over land are low, and furthermore most of the ATom data were over water. \*\*

- In subsection 3.2 the sea salt correlation with humidity is taken as indicator for wet removal of the particles. In contrast, on page 9 the correlation is explained as indicator for mixing of air masses. Please clarify.

\*\*The mixing is with air after wet removal in convection so the distinction is about when the wet removal occurred. The sentence after "If such intense convective clouds scavenge nearly all sea-salt particles…"has been reworded to emphasize removal. \*\*

- The authors refer to model results shown in Figure 7 partly from submitted or 'in preparation' publications. This should be avoided.

\*\*The publication in question has now been accepted and the reference updated. The Bian et al. paper is now in discussion so the text references to it have also been updated (as still in discussion, it is not a formal reference). \*\*

\*\*Besides the review responses, we have made minor grammatical changes and moved a sentence on page 10 (right before section 3.1) to the previous paragraph where it fits better. \*\*

---

## Author Response (AR2)

Thank you for your help. Besides adding units to Figure 2 we added one funding acknowledgment.